# Is Feature Extraction the most informative dimensionality reduction technique? Revisiting Unsupervised Feature Selection from a Dynamic Approach

## Abstract

This paper compares unsupervised feature extraction and unsupervised feature selection techniques in the context of dimensionality reduction without using labeled data. Unsupervised feature extraction transforms the input space into a lower-dimensional representation by creating informative features that capture underlying patterns, leading to improved model performance. On the other hand, unsupervised feature selection chooses a subset of features based on predefined criteria, potentially overlooking important relationships and reducing the model's discriminative power. State-of-the-art researches suggest that feature extraction outperforms feature selection in terms of model accuracy and robustness. Leveraging the intrinsic structure of the data, unsupervised feature extraction provides richer representations, enhancing the model's ability to discern complex patterns. These paper proposes to revisit feature selection algorithms from a dynamic perspective, where the features are selected depending on the specific sample input. Through empirical evaluations, it will be demonstrated that unsupervised feature selection outperforms feature extraction, both in accuracy and data compression. These findings highlight the potential of unsupervised feature selection as a powerful approach for dimensionality reduction and improved model performance, particularly when labeled data is scarce or unavailable.

## 1 Introduction

Feature selection and feature extraction are two essential techniques in the field of machine learning and data analysis, both aimed at improving the efficiency and effectiveness of predictive modeling Zebari et al. (2020). While both approaches share the goal of reducing the dimensionality of datasets, they diverge significantly in their fundamental strategies and objectives.

Feature selection goal is to identify a subset of relevant features from a larger set of available features that are most informative for a given task Dhal & Azad (2022). Traditional feature selection methods typically rely on static criteria, selecting a fixed set of features prior to model training Balın et al. (2019); Roffo et al. (2020); Cancela et al. (2023). However, in dynamic and evolving data environments, where the relevance of features may change over different samples, static feature selection may not be optimal. Dynamic Feature Selection (DFS) (also referred as *Instance-based or Instancewise Feature Selection* Yoon et al. (2018); Panda et al. (2021); Liyanage et al. (2021)) represents a paradigm shift in feature selection by recognizing the variability of feature importance Chen et al. (2018); Arik & Pfister (2021) between each sample. Unlike traditional static methods, DFS algorithms adaptively adjust the feature subset during model training or deployment, accommodating changes in data characteristics and task requirements.

Moreover, feature extraction techniques transform the original feature space into a new space by creating a set of derived features that capture essential information from the original data. These methods, such as Principal Component Analysis (PCA), Linear Discriminant Analysis (LDA) or any deep features derived from a deep learning model, seek to maximize the discriminative power or variance of the transformed features Perera & Patel (2019); Tang et al. (2022); Izmailov et al.

(2022). In doing so, they often create a smaller, more compact representation of the data, potentially enhancing model performance, in spite of its interpretability.

While both feature extraction and feature selection serve to enhance the quality of input data for machine learning models, they differ fundamentally in their approach. Feature extraction generates entirely new features, potentially altering the interpretability of the data, while feature selection retains the original features, preserving the original meaning and context. In this context, DFS arises as a technique that tries to merge the versatility of feature extraction techniques with the interpretability of feature selection approaches.

In this paper, a comprehensive study over the properties of Dynamic Feature Selection is conducted, elucidating its fundamental principles, methodologies, and key differences compared to traditional feature extraction techniques. A novel DFS method will be presented, aiming to provide an alternative to the classic data representation using deep features.

The main contributions of this paper are the following:

- To our knowledge, this is the first attempt to provide a DFS framework for unsupervised scenarios.
- A novel dynamic feature selection method, called Dynamic Data Selection (DDS) will be presented. Contrary to previous approaches like Chen et al. (2018), the memory consumption of this approach will be minimal, and invariant to the maximum number of selected features.
- A variation to the hard concrete distribution Louizos et al. (2018) will be presented. Taking the properties of the DDS algorithm, it will be used during the training procedure to minimize the chances of premature feature eliminations.
- The algorithm will be tested in two different unsupervised scenarios, data compression and clustering, to show its adaptability to different scopes, as well as its ease of use. The DDS model allows the architect designer to use more complex networks, since it preserves the position of the selected features.

This paper is organized as follows: section 2 will introduce the most important DFS approaches; section 3 will describe the problem formulation, along with some details regarding the implementation and the training procedure; section 4 will introduce some experimental results about the ability of this dynamic feature selection approach to both compress the input data, and to how it can successfully be used in a different task, like clustering; and finally, section 5 will provide some conclusions and future work.

## 2 RELATED WORK

DFS is a very recent field of study, as almost no contributions were provided prior to the rise of deep learning architectures. Three works stand out among all: *Learning to Explain* (L2X) Chen et al. (2018), *INVASE* Yoon et al. (2018) and TabNet Arik & Pfister (2021). However, it is worth noting that these algorithms were entirely developed for supervised learning.

L2X Chen et al. (2018) presented an autoencoder-like architecture that is attached prior to the classification model. The output of the architecture is of the form $\mathcal{R}^{N \times F \times M}$, being $M$ the maximum number of features to be selected. Then, each input sample is transformed by performing a matrix multiplication. Although this solution provides remarkable results in supervised scenarios, it suffers from two major drawbacks: first, the memory requirement, as the output size of the model is dependent on the number of maximum features to be selected, forcing $M$ to have smaller values; the second drawback is related to the matrix multiplication procedure, as it forces to input data to be one dimensional, disabling the classification model to have complex layers like 2D convolutions.

INVASE Yoon et al. (2018) consists on 3 different networks, a selector, a predictor and a baseline. Inspired by the actor-critic method Peters & Schaal (2008), the predictor and the baseline will output the classification scores, only differing in the input data: while the baseline introduced the whole features, the predictor will only select as input a small subset of the features. This subset will be chosen by the selector algorithm. Compared to L2X, the main advantage of this algorithm is that is able to preserve the spatial information of the input data. As major drawbacks, it is very challenging

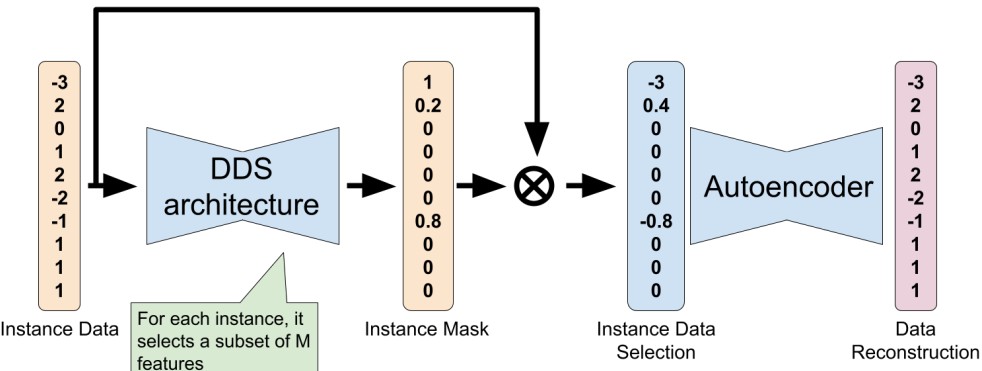

Figure 1: Model architecture. The Dynamic Data Selection model will be in charge of selecting, per sample, a fixed subset of features. Those features will solve the unsupervised task in hand (in this particular example, data reconstruction).

to successfully train all 3 networks at the same time. It will be shown that the proposed DDS method can embed this idea by using only one network and a training trick (see section 3.2.2).

Moreover, TabNet Arik & Pfister (2021) uses sequential attention to select the most important features per sample. It was specially designed to be used in text problems. The main advantage of this approach is the inherent explainability, as it is easy to depict the reasoning behind the classification decision. On the other hand, its accuracy is lower than L2X and INVASE.

Although recent contributions on DFS were conducted Shah et al. (2022); Covert et al. (2023), they are based on the models explained before.

## 3 PROBLEM FORMULATION

Let $\mathbf{X} \in \mathcal{R}^{N \times F}$ be our input data, where $N$ is the number of instances and $F$ is the total number of different features. Our aim is to select, for each instance, a maximum number of features, denoted by $M$. Formally speaking, our algorithm aims to solve the following minimization problem:

$$
\begin{aligned}
\underset{\boldsymbol{\Theta}_{DDS}, \boldsymbol{\Theta}_{UNSUP}}{\text{minimize}} \quad & \mathcal{L}(f(g(\mathbf{X}; \boldsymbol{\Theta}_{DDS}) \circ \mathbf{X}; \boldsymbol{\Theta}_{UNSUP})) \\
\text{subject to} \quad & g(\mathbf{X}; \boldsymbol{\Theta}_{DDS}) \in [0, 1]^{N \times F}, \\
& \|g(\mathbf{X}; \boldsymbol{\Theta}_{DDS})^{(i)}\|_0 \leq M, \forall i \in \{1..N\}.
\end{aligned}
\tag{1}
$$

where $\mathcal{L}$ is the unsupervised loss function, $M$ is the maximum number of features to be selected, $g(\cdot; \boldsymbol{\Theta}_{DDS})$ is the DDS network, and $f(\cdot; \boldsymbol{\Theta}_{UNSUP})$ is the unsupervised task one may one to solve. For the sake of simplicity, initially $f(\cdot; \boldsymbol{\Theta}_{UNSUP})$ will be considered as an autoencoder that aims to reconstruct the initial features, including those that are masked by $g(\cdot; \boldsymbol{\Theta}_{DDS})$. In Section 4 it will be shown how one can successfully change it to solve use other unsupervised tasks. The key idea is simple: an autoencoder architecture ($f$) is trained while introducing an extra network ($g$), which will be in charge of selecting, at most, $M$ relevant features per sample. After that, those features will be passed to the autoencoder model, aiming to reconstruct the whole input data.

In this paper, a novel DDS algorithm is proposed, where the output of the model is $\mathcal{R}^{N \times F}$, that is, it is not dependent on the maximum number of selected features. Note that this method is very easy to use, as it is defined to be a module to be inserted prior to the main task. The input of the principal model will be the input data masked by the output of the DDS module. Thus, it can be easily adapted to be included in any task that can be trained by using gradient minimization.

## 3.1 DDS IMPLEMENTATION

As depicted in Eq. 1, two constraints have to be addressed to properly solve the minimization problem. Although this formulation is very similar to a recent feature selection algorithm called E2E-FS Cancela et al. (2023), the solution proposed by the authors cannot be applied in this particular dynamic approach. E2E-FS aims to select the same $M$ features for all instances, forcing the discarded ones to be zeroed. Although this is a good solution for a FS approach, it is too restrictive in a dynamic version. We aim to keep the scores of the discarded features, so they can be used for other purpose, like model explanation. Thus, we propose to reformulate the problem as

$$
\begin{aligned}
&\underset{\boldsymbol{\Theta}_{DDS}, \boldsymbol{\Theta}_{UNSUP}}{\text{minimize}} && \mathcal{L}(f(\tau(\tilde{g}(\mathbf{X}; \boldsymbol{\Theta}_{DDS})) \circ \boldsymbol{\Gamma}_M \circ \mathbf{X}; \boldsymbol{\Theta}_{UNSUP})) + \alpha \mathcal{L}_0(\tilde{g}(\mathbf{X}; \boldsymbol{\Theta}_{DDS})) \\
&\text{subject to} && \tau(\tilde{g}(\mathbf{X}; \boldsymbol{\Theta}_{DDS})) \in [0,1]^{N \times F}, \\
& && \boldsymbol{\Gamma}_M \in \{0,1\}^{N \times F}, \\
& && \|\boldsymbol{\Gamma}_M\|_0 \le M, \forall i \in \{1..N\}.
\end{aligned}
\tag{2}
$$

where

$$
\tau(x) = \min\left(1, \max\left(0, \sigma\left(\frac{x}{\beta}\right)(\zeta - \gamma) + \gamma\right)\right),
\tag{3}
$$

begin $\sigma$ the sigmoid function. $\tau(x)$ is the hard concrete gate presented in Louizos et al. (2018).

The idea is to split the DDS model $g$ into two different terms. In first place, $\tilde{g}(\cdot; \boldsymbol{\Theta}_{DDS})$ is implemented in the same way as $g$ was previously defined, without the 0-norm constraint. This constraint is now placed to a different matrix, called $\boldsymbol{\Gamma}_M$. $\boldsymbol{\Gamma}_M$ is a binary matrix with all zeroes but the top-M scores of each sample of $\tilde{g}(\cdot; \boldsymbol{\Theta}_{DDS})$. Mathematically, each row $\boldsymbol{\Gamma}_M^{(i)}$ can be recursively defined as

$$
\begin{aligned}
\mathbb{T}_0^{(i)} &= \emptyset \\
\mathbb{T}_{n+1}^{(i)} &= \mathbb{T}_{n+1}^{(i)} \cup \{\tilde{g}(\mathbf{X}; \boldsymbol{\Theta}_{DDS})^{(i)} \setminus \mathbb{T}_n^{(i)}\} \\
\boldsymbol{\Gamma}_M^{(i)} &= \begin{cases} 1 & \forall f \in \mathbb{T}_M^{(i)} \\ 0 & \text{otherwise} \end{cases}
\end{aligned}
\tag{4}
$$

Besides that, it is still required to add a regularization to the output of $\tilde{g}$, aiming to force to have low importance values only to the less informative features. This is done by the $\mathcal{L}_0$ term, which is defined as the $L_0$ regularization term presented in Louizos et al. (2018):

$$
\mathcal{L}_0(x) = \frac{1}{N} \sum \sigma\left(x - \beta \log \frac{-\gamma}{\zeta}\right)
\tag{5}
$$

During all the experiments, the hyper-parameters where set to $\alpha = 2 \cdot 10^{-5}$, $\beta = \frac{2}{3}$, $\zeta = 1.1$ and $\gamma = -0.1$.

## 3.2 TRAINING PROCEDURE

The training procedure is similar to the one used in Louizos et al. (2018), with two main differences:

- A variation of the hard concrete distribution Maddison et al. (2016); Jang et al. (2016) presented in Louizos et al. (2018) is created, aiming to increase the probability mass of values close to 1.
- Two training tricks with decaying factor are used, in order to prevent the weight initialization to play an important role in the final result.

### 3.2.1 VARIATION TO THE HARD CONCRETE DISTRIBUTION

The hard concrete distribution presented in Louizos et al. (2018) aimed to increase the probability mass near 0 and 1, in order to either force some features to be discarded, or to increase the feature probability to near 1. In our case, the discard part in not necessary, as the $\boldsymbol{\Gamma}_M$ matrix will already

discard all the features that are not part of the top-M most important features for each instance. Is fact, using the original hard concrete distribution is also counterproductive, as it could unadvisedly force the model to remove more features than the target $M$, causing a degrade in the performance of the algorithm.

In order to avoid this problem, a variation of this distribution is presented, aiming to only increase the probability mass only near 1. This distribution is defined as

$$\tau_u(x) = \min\left(1, \max\left(0, \sigma\left(\frac{x - 2\log(u)}{\beta}\right)(\zeta - \gamma) + \gamma\right)\right), \quad u \in \mathcal{U}(0, 1). \tag{6}$$

These distribution will ensure that, at some point, all values in $\mathbf{\Gamma}_M$ will be 1 at some point of the training procedure, without causing any undesired removal.

### 3.2.2 Decaying factor for training tricks

Although the inclusion of Eq. 6 during training helps the stability of the procedure, it also causes a slight under-fitting over the training of the original problem $f(\cdot; \mathbf{\Theta}_{UNSUP})$. Initial experiments conducted for this study showed that Eq. 6 help is focused on the beginning of the training. In order to avoid the under-fitting problem over $f(\cdot; \mathbf{\Theta}_{UNSUP})$, a decay factor is placed over Eq. 6, aiming to erase $\tau_u(x)$ over the last steps of the training procedure. The modified version is defined as

$$\tilde{\tau}(x) = \alpha_\tau^t \tau_u(x) + (1 - \alpha_\tau^t)\tau(x) \tag{7}$$

being $t$ the training iteration. During the experiments, $\alpha_\tau$ was set to 0.99995.

Besides that, it was also noted that the model initialization affects the training result by a huge margin. Following Cancela et al. (2023), it was found that $f(\cdot; \mathbf{\Theta}_{UNSUP})$ should be properly warmed-up to provide a consistent training. However, it is possible to circumvent it by substituting, with a low probability, $\mathbf{\Gamma}_M$ for $\mathbf{\Gamma}_F$, that is, selecting all features. In a similar fashion with the previous trick, this probability will decay until it disappears. Strictly speaking, $\mathbf{\Gamma}_M$ is substituted by

$$\mathbf{\Gamma} = \begin{cases} \mathbf{\Gamma}_F & \text{with probability} \quad p = \min(\epsilon_\Gamma, \alpha_\Gamma^t) \\ \mathbf{\Gamma}_M & \text{with probability} \quad p = 1 - \min(\epsilon_\Gamma, \alpha_\Gamma^t) \end{cases} \tag{8}$$

being $t$ the training iteration. During the experiments, $\alpha_\Gamma$ was set to 0.9995 and $\epsilon_\Gamma$ was set to 0.2. These two tricks will help the training procedure to be invariant to the model initialization.

## 4 Experiments

The experimental section of this paper will be divided in two main tasks, aiming to solve two different questions. First, some experiments regarding to data compression will be conducted, aiming to test if this dynamic feature selection technique could be considered as a replacement of a classic feature extraction method. Finally, the DDS model will be attached to a state-of-the-art clustering technique, in order to show the versatility of this approach. All algorithms, scripts and results are accessible via GitHub[1]

### 4.1 Data Compression

In order to evaluate the ability of the DDS algorithm to reconstruct the input data, two different image datasets have been selected: MNIST LeCun et al. (1998), composed of $28 \times 28$ gray-scale images (almost binary); and CIFAR-10 Krizhevsky et al. (2009), composed of $32 \times 32$ RGB images.

As stated in section 1, one of the main advantages of the DDS algorithm is that it is able to preserve the spatial location of the selected features. This means that the output of the DDS procedure has the same shape of its input. In a data reconstruction environment, this effect causes a huge advantage, because more complex networks that a classic autoencoder can be used to reconstruct the input data from the DDS output.

---

[1]URL will be included upon acceptance.

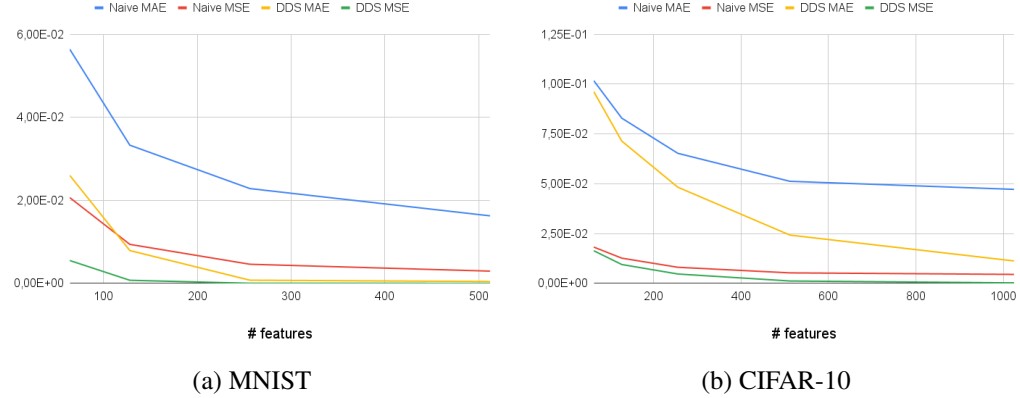

(a) MNIST                                    (b) CIFAR-10

Figure 2: Reconstruction errors when using the same number of features in both architectures. The DDS model is able to obtain a more accurate representation of the input data in both datasets.

In this scenario, the U-Net architecture Ronneberger et al. (2015) was selected to model either the DDS architecture $g(\cdot; \boldsymbol{\Theta}_{DDS})$ and the reconstruction part $f(\cdot; \boldsymbol{\Theta}_{UNSUP})$. It consists of a fully convolutional network with a contracting and an expansive path. Starting with $C$ channels in the first layer, the number of channels is doubled every time a 2-downsampling procedure is performed. Multiple residual links connect both the contracting and the expanding layers.

In order to make a fair comparison against a similar network, a naive U-Net model without the residual links was selected. As a classic autoencoder, the output of the deepest layer will be considered as the extracted deep features. The number of deep features will depend on the number of initial channels $C$. Aiming also to remove the network size as a cause of the disparity of the results, the number of channels in the naive network will be doubled with respect to each U-Net in the DDS configuration. With this simple modification, a similar number of trainable weights will be ensured.

As loss function, a elastic approach was selected, defined as

$$\mathcal{L}(\mathbf{X}, \tilde{\mathbf{X}}) = \|\mathbf{X} - \tilde{\mathbf{X}}\|_2^2 + \alpha_L \|\mathbf{X} - \tilde{\mathbf{X}}\|_1, \tag{9}$$

being $\mathbf{X}$ the input data, $\tilde{\mathbf{X}}$ the data reconstruction, and $\alpha_L = 10^{-2}$ the hyper-parameter controlling the L$_1$ norm. The Adam optimizer Kingma & Ba (2014) with a learning rate of $10^{-3}$ was used for 100 epochs, with a batch size of 256. The algorithms were trained over the train partitions of each dataset, whereas the results provided in this paper correspond to the test partition.

This experiment aims to answer two different questions:

- Can dynamic feature selection be an alternative to feature extraction whenever they share the same number of features?

- Can dynamic feature selection be an alternative to feature extraction whenever their memory consumption is similar? Although dynamic feature selection has the advantage of preserving the input data positioning, it also requires to allocate an extra memory space to store the index of the selected positions.

### 4.1.1 DFS VS FE: USING THE SAME NUMBER OF FEATURES

In this experiment, DDS will select the same number of features than the number of deep features provided by the naive model. By controlling the number $C$ of channels of the first layer, we can modify the number of deep features. Fig. 2 shows both the Mean Average Error (MAE) and the Mean Squared Error (MSE) of both models, over the two datasets. The experimental results conducted over the MNIST dataset clearly indicate that the DDS Algorithm significantly outperformed the naive autoencoder. It also have a MAE lower than the MSE of the naive model, which is impressive. It suggest that the DDS network is very capable of reconstructing data that almost only contain two different values (0s and 1s).

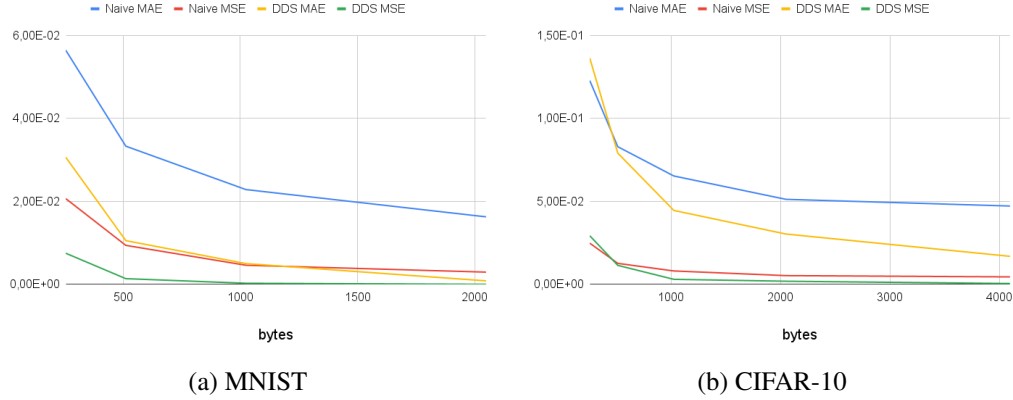

(a) MNIST

(b) CIFAR-10

Figure 3: Reconstruction errors when using the same storing memory in both architectures. Besides when selecting a small number of featues in the CIFAR-10 dataset, the DDS model is able to obtain a more accurate representation of the input data.

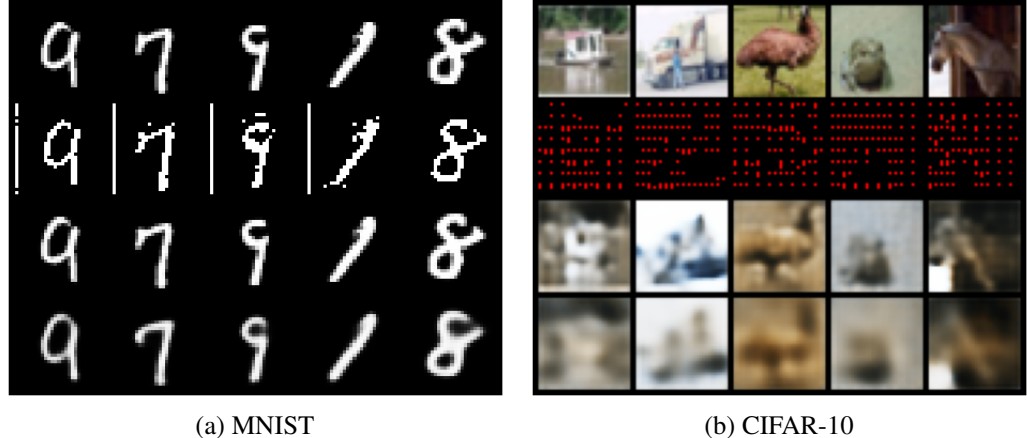

(a) MNIST

(b) CIFAR-10

Figure 4: Reconstruction examples for both datasets when using the same amount of bytes to store the data compression (equivalent to 128 features for the naive model). From top to bottom: original image, DDS selected features (white in MNIST, red in CIFAR-10), DDS reconstruction and Naive reconstruction.

On the contrary, the results obtained in CIFAR-10 are closer between the two models, specially when the number of features is small. Nevertheless, an interesting effect arises when looking at the MAE scores: as the number of features increases, the MAE improvement in the naive model quickly decreases, showing almost no gain when the number of features is higher than 500. However, this bottleneck does not affect to the DDS model, as the MAE continues to reduce, increasing the error gap between these two approaches.

### 4.1.2 DFS VS FE: USING THE SAME MEMORY ALLOCATION

As early mentioned, if the main purpose of the task is to store the selected features, it is also needed to save some space for their indexes. Thus, this test will show the quality of the data reconstruction when both models use the same space to store their data compression information. Knowing that each MNIST image has 784 pixels, and the CIFAR-10 images contain 3072 features, 10 and 12 bits will be needed to store each index position, respectively. For instance, let the naive model outputs 128 32-bit float point deep features for the MNIST dataset. In order to occupy the same amount of memory, the DDS model can only select 97 features.

Fig. 3 shows the obtained results. Again, the results obtained by the DDS model overcome the nave model results by a huge margin, suggesting that dynamic feature selection could be a better data

Table 1: Comparative clustering results on four different datasets. The DDS model is able to maintain similar results to thos obtained by the algorithm attached to it, even when using a low number of selected features.

| MODEL | CIFAR-10 | | | CIFAR-100/20 | | | STL-10 | | | TINY-IMAGENET/200 | | |
|---|---|---|---|---|---|---|---|---|---|---|---|---|
| | ACC | NMI | ARI | ACC | NMI | ARI | ACC | NMI | ARI | ACC | NMI | ARI |
| DAC | 52.2 | 39.6 | 30.6 | 23.8 | 18.5 | 8.8 | 47.0 | 36.6 | 25.7 | 6.6 | 19.0 | 1.7 |
| DCCM | 62.3 | 49.6 | 40.8 | 32.7 | 28.5 | 17.3 | 48.2 | 37.6 | 26.2 | 10.8 | 22.4 | 3.8 |
| SCAN | 88.3 | 79.7 | 77.2 | 50.7 | 48.6 | 33.3 | 80.9 | 69.8 | 64.6 | - | - | - |
| CC | 79.0 | 70.5 | 63.7 | 42.9 | 43.1 | 26.6 | 85.0 | 76.4 | 72.6 | 14.0 | 34.0 | 7.1 |
| TCL | 88.7 | 81.9 | 78.0 | 53.1 | 52.9 | 35.7 | 86.8 | 79.9 | 75.7 | - | - | - |
| IMC-SwAV | 89.1 | 81.1 | 79.0 | 49.0 | 50.3 | 33.7 | 83.1 | 72.9 | 68.5 | 27.9 | 48.5 | 14.3 |
| DDS(10%)+ IMC-SwAV | 82.0 | 70.4 | 66.7 | 39.4 | 40.9 | 25.3 | 82.7 | 72.2 | 67.5 | 23.3 | 49.2 | 11.3 |
| DDS(25%)+ IMC-SwAV | 85.9 | 76.6 | 73.4 | 45.0 | 47.0 | 29.9 | 84.8 | 74.4 | 70.7 | 26.4 | 51.3 | 13.3 |
| DDS(50%)+ IMC-SwAV | 89.1 | 80.5 | 78.9 | 48.9 | 49.8 | 33.5 | 84.8 | 74.4 | 70.8 | 26.5 | 51.3 | 13.3 |
| DDS(100%)+ IMC-SwAV | 89.3 | 81.1 | 79.4 | 51.2 | 52.1 | 35.3 | 84.5 | 73.6 | 70.2 | 27.8 | 52.5 | 14.2 |

compression protocol for this special type of dataset. With respect to the CIFAR-10 dataset, the naive model is able to obtain a better data compression when using a very low number of features (64). However, as the number of features is increased, the data compression quality of the DDS model surpasses the naive model by a huge margin. Fig. 4 also shows some random examples of data reconstruction on both datasets. The DDS reconstruction provides a less blurry solution than the one offered by the naive model. Note that, although the algorithm selects some border pixels in the MNIST dataset, their importance score is zero. In practice, it is like these pixels were not selected.

## 4.2 CLUSTERING

An extra experiment will be conducted, aiming to test the ability of the DDS model to adapt it to different unsupervised scenarios. This experiment will show the results obtained in a clustering problem, when the DDS model is just attached to a state-of-the-art architecture, without performing any calibration or hyper-parameter tuning.

Over the recent years, the contrastive learning boosts the quality of the unsupervised clustering in images. Techniques like Contrastive Clustering (CC) Li et al. (2021) or its upgrade, the Twin Contrastive Clustering (TCL) Li et al. (2022) achieved results near to those obtained by supervised techniques, which is remarkable. However, some of these works perform image resizing, changing the initial CIFAR-10 image size $32 \times 32$ to a much bigger one ($224 \times 224$ for TCL, for instance). Therefore, it makes no sense to perform a dynamic feature selection over an image that is artificially enlarged.

In order to make a fair experiment, DDS should be attached to a clustering model that does not require image resizing to achieve state-of-the-art results. Over these solutions, IMC-SwAV Ntelemis et al. (2022) stands out from the rest. Therefore, the DDS module $g(\cdot; \Theta_{DDS})$ was attached to the IMC-SwAV model $f(\cdot; \Theta_{UNSUP})$, without incurring in any optimization. The same training procedure presented in Ntelemis et al. (2022) will be applied here. The U-Net model (with $C = 16$) will be used as the DDS architecture.

As baselines, in addition to the aforementioned CC, TCL and IMC-SwAV, models like DAC Chang et al. (2017), DCCM Wu et al. (2019) and SCAN Van Gansbeke et al. (2020) were also included. Table 1 shows the clustering results obtained over four different datasets: CIFAR-10, CIFAR-100/20 Krizhevsky et al. (2009), STL-10 Coates et al. (2011) and Tiny-ImageNet/200 Le & Yang (2015), which is a subset of ImageNet containing 200 classes downsampled to a lower resolution. The results show that DDS is able to significantly reduce the number of input features without incurring

in a huge increase of the IMC-SwAV performance. Besides that, it is also able to keep or even increase its performance when selecting at least half of the input features.

It is specially interesting to note that, even when only selecting the 10% of the input features, the score obtained is higher than state-of-the-art techniques of 4 years ago. These results suggest that DDS can be attached to any state-of-the-art model with almost no clustering performance loss.

## 5 DISCUSSION

A general recipe for Dynamic Feature Selection on unsupervised scenarios was presented in this paper, that allows to attach the so-called DDS architecture to the input of any unsupervised task. The method is based on the creation of an autoencoder-like architecture that outputs the selection of, at most, $M$ relevant features with its respective score, being $M$ a fixed parameter tuned by the operator. This output is regularized by an $L_0$ regularization, ensuring that unimportant features will receive close to zero importance. A novel concrete distribution was created, aiming to introduce variability in the dynamic feature selection without forcing any of them to be prematurely removed. In the experiments, it was shown that the DDS architecture can perform data compression with better results than a classic autoencoder, even when taking into account that extra memory is needed for saving the feature selection indexes. This effect is caused by two factor: first, the information provided by the selected features is extremely discriminative; and second, more complex autoencoder-like architectures, like U-Net, can be used with this approach, since the input data structure is always preserved. It was also shown that the DDS architecture can be attached to a state-of-the-art clustering model, achieving similar results even when selecting a tiny fraction of the input data.

It is worth noting that the output of the DDS architecture is not forced to be binary. In fact, preliminary studies show that the feature importance score barely reaches the perfect score of 1. This could be a problem in terms of explainability, a the stored compressed data is modified from the original one. Thus, two solutions can be performed to solve this problem. In first place, it is possible to store the input data and their importance score in separate ways, but it causes the memory requirements to be almost doubled. The second solution is to force the DDS output to be binary by introducing more restrictions into the model. Although initial tests show a remarkable degrade in the DDS performance, only a small fraction of techniques were used to force this extra restriction.

For future work, it is planned to further study the issue mentioned before. It would be also interesting to take advance of the ability of the DDS architecture to preserve the input data structure. Novel contrastive learning loss functions can be derived from this idea, as the selected pixels of an image should be similar no matter how many geometrical operations are applied to perform data augmentation over them. Finally, it would also be interesting to extend this architecture to the supervised scenario, in a similar fashion previous DFS algorithms did.

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
