# OpenReview forum: "Is Feature Extraction the most informative dimensionality reduction technique? Revisiting Unsupervised Feature Selection from a Dynamic Approach"
_ICLR.cc/2024/Conference — Submitted to ICLR 2024_

### Official Review · Reviewer_LSnn · 2023-10-27

**Soundness:** 2 fair
**Presentation:** 2 fair
**Contribution:** 2 fair
**Rating:** 5
**Confidence:** 2

**Summary:**

A method for dynamic unsupervised feature selection is proposed and compared against feature extraction methods. Experiments on real data demonstrate improve performance over the baseline model on two real world data sets, MNIST and CIFAR-10.

**Strengths:**

The problem of dynamic feature selection in an unsupervised context is novel and potentially very impactful. Spatial information is preserved allowing easy interpretation of the selected features.

**Weaknesses:**

The paper is quite complex and the writing is unclear in places. The technical description of the method is hard to follow, particularly the equations surrounding (4): $\mathbb{T}^{(i)}_{n+1}$ here seems to be defined in terms of itself, and $f$ in the equation just below it is unused. There are no experiments on simulated data where important features are known are present, or theoretical guarantees, leaving it unclear as to whether this method can recover the correct features.

**Questions:**

- Figure 4 a shows the selected features, but these look a bit strange as it appears the left hand border of the image is selected. This doesn't seem right as those pixels are all black and have little variance. What's causing these to be selected?
- It is written that "This output is regularized by an L0 regularization, ensuring that unimportant features will receive
close to zero importance." -> is there any guarantees that the correct features are recovered? $L_0$ regularisation by itself does not guarantee this.
- Section 3.2.1 introduces an alternative distribution but I didn't see where the effects of this change were evaluated. Have I missed it?
- There are many hyperparameters such as various decay rates like $\alpha_\tau$ and $\alpha_\Gamma$. How stable are the solutions?

---

### Official Review · Reviewer_eFwU · 2023-10-30

**Soundness:** 2 fair
**Presentation:** 2 fair
**Contribution:** 2 fair
**Rating:** 5
**Confidence:** 4

**Summary:**

The paper introduces the DDS architecture, a dynamic feature selection method tailored to instance-specific feature selection. This approach reimagines the traditional paradigms of feature selection by focusing on individual data instances.

Results highlight the potential of unsupervised feature selection in enhancing dimensionality reduction and the models' performance. Through empirical evaluations, the research shows the effectiveness of their architecture, especially in scenarios where labeled data is scarce or unavailable.

**Strengths:**

The paper introduces a method for a dynamic feature selection algorithm, where the model could offer flexibility by allowing features to be dynamically selected. The paper is well-written, and the authors conducted experiments to cover data comparison and clustering, showing the model performance. The structure of the paper is logical, guiding readers through its key contributions.

**Weaknesses:**

The paper introduces a method for dynamic feature selection, but there's a lack of detail regarding the reason for choosing parameters, or statistical measures adopted, limiting the reproducibility of the work.
It would be beneficial to provide more depth into the selection features to describe the results in the MNIST dataset and visualization of more samples from other digits (3, 4, 5, ... ).
In section 4.1 while the experimental results on the MNIST dataset are promising, it's important to note that MNIST is notably sparse. This might particularly favor the DDS Algorithm's performance. Additionally, the vertical lines (representing selected features) in the second row of Figure 4 might indicate potential pitfalls, such as the selection of non-informative features.

Every feature selection algorithm comes with its limitations in finding the optimal set of features. The paper could benefit from discussing the potential cases where their method might not be suitable.

**Questions:**

1. Please add the y-axis labels for Figure 1 and Figure 3.
2. Considering the sparsity of the MNIST dataset, does it inherently enhance the DDS Algorithm's performance in Figure 1(a) and Figure 4(a)? Kindly elaborate.
3. Could you clarify the reasoning behind the selection of the vertical lines (features) in the second row of Figure 4 for sample MNIST images?
4. How does the DDS Algorithm ensure it avoids selecting non-informative features?
5. How did you determine the parameter values detailed in sections 3.1 and 3.2.2?
6. Are there any limitations when applying the DDS algorithm?

---

### Official Review · Reviewer_rUoV · 2023-10-30

**Soundness:** 2 fair
**Presentation:** 2 fair
**Contribution:** 3 good
**Rating:** 5
**Confidence:** 4

**Summary:**

Authors of this paper propose a novel framework for dynamic feature selection (DFS) where a sample-based DDS network is used to learn feature weights for a given sample and indicator matrix is used to remove features. Two tricks are proposed to stabilize the training so that it is not sensitive to the weight initialization.

**Strengths:**

A DDS network is used to select features for each sample, which is jointly optimized with task-related model such as autoencoder. The separation of weighing and indicator matrices are introduced. Two tricks are used to stabilize the training so that it is not sensitive to the weight initialization.

**Weaknesses:**

Some important parts of the proposed model need to be clarified including some error in recursive update for the binary matrix, equation (1) and the contradictory explanation on how tau(x) should be used. The description of Abstract is quite deviated from the main context of this work. The comprehensive study on DFS claimed in Section 1 is missing. Experiments can be improved by including more proper baselines and detailed experimental settings.

**Questions:**

It seems that the abstract of this paper does not present the proposed work properly.

The comprehensive study over DFS is not shown except the brief discussion of three methods in related work.

The equation (4) is not a recursive function from mathematical perspective. It is hard to understand (4) since tilde{g} is not a set.

The equation (6) needs further explanation on why it does not cause any undesired removal because the additional term log(u) where u from uniform distribution is used.

It is controversial that (6) is introduced because (3) can cause degradation due to unadvisedly forcing the model to remove features, while in (7) it is included at the later stage of the training.

When the autoencoder is used in (1), the aim is to reconstruct the whole input data. It means that a set of selected features for each sample must recover the whole sample. Can this recovery be reasonable, especially for the case where many features are noisy?

From Table 1, TCL works better than IMC-SwAV, which shows different observations as discussed in section 4.2.

In Figure 2 and Figure 3, authors leverage U-Net architecture to implement DDS where multiple residual links are used, but the baseline model as the naïve U-Net without residual links. Is it possible that the results reported may be caused by the residual links instead of the DDS selection module?

DDS with 10% selected features suffers significant degradation comparing with DDS with full set of features and IMC-SwAV. Since DDS is integrated into IMC-SwAV, it is unfair to compare with the models of 4 years ago. Hence, for IMC-SwAV, DDS with few features do hurt the clustering performance.

---

### Official Review · Reviewer_tqPc · 2023-11-01

**Soundness:** 2 fair
**Presentation:** 1 poor
**Contribution:** 1 poor
**Rating:** 3
**Confidence:** 4

**Summary:**

The authors provide an interesting comparison between unsupervised feature selection and extraction. Feature selection and extraction can both be used to find compressed representations of data. The first works by transforming input features into a low-dimensional representation, and the latter seeks subsets of informative features.

Besides this comparison, they propose an unsupervised dynamic feature selection method.
The idea is to use a NN to predict for each sample which features are useful for reconstruction via a reconstruction loss.

**Strengths:**

Both feature selection and feature extraction are useful tools with numerous applications. The comparison of these two provides an interesting viewpoint that could be valuable for practitioners. The proposed method also seems valuable for compression and clustering.

**Weaknesses:**

The paper is not well written; many parts are unclear, and the many phrasings and descriptions need to be improved.
The concept of dynamic (or local) feature selection was previously studied in the context of supervised learning. The authors mention L2X and INVASE which are designed for interpretability, but do not mention LSPIN [1]. Since the differences are subtle, the method should be compared to LSPIN with an AE architecture (that’s not a big difference; it only requires changing the architecture and using a regression loss MSE).
The approach includes many tricks that are not well explained or motivated. Also, they do not provide an ablation study or sensitivity analysis to provide intuition about the usefulness of these tricks.
My biggest concern is that both feature selection and feature extraction are typically used for high-dimensional data. In particular, feature selection is generally used in the underdetermined regime (N<D), or in the presence of nuisance features. Moreover, images typically don’t have nuisance features and are less appropriate for FS since a specific pixel doesn’t have a universal meaning, and there is not much value in sampling fewer pixels from a practical point of view (cameras are cheap and already implemented). Often, MNIST is used for demonstration purposes, mostly because the digits are centered.
So, the authors are missing an evaluation of the method on tabular data, which is more appropriate for feature selection. I’m also missing an example demonstrating the feature selection capabilities of the method, namely a controlled environment in which the authors can show that the method identifies informative features and attenuates nuisance features.

[1] Yang, et al.. "Locally sparse neural networks for tabular biomedical data." International Conference on Machine Learning. PMLR, 2022.

**Questions:**

In the abstract: “state of the art researches” is that a proper phrasing? Maybe the method is SOTA, but not the research

Instance wise FS, missing citation of [1]
[1] Yang, et al.. "Locally sparse neural networks for tabular biomedical data." International Conference on Machine Learning. PMLR, 2022.

“To the best of our knowledge, this is the first attempt to provide a DFS for unsupervised scenarios”

This is incorrect [2] provides an unsupervised FS that is dynamic (although the final goal is classification, it could also be used for clustering)

[3] provides a method for clustering with dynamic FS, completely unsupervised.

[2] Nunes, Rômulo de O., et al. "An unsupervised-based dynamic feature selection for classification tasks." 2016 International Joint Conference on Neural Networks (IJCNN). IEEE, 2016.
[3] Svirsky, Jonathan, et al. "Interpretable Deep Clustering." arXiv (2023).

Why is most of the paper written in future tense? For example:
“The algorithm will be tested”... No, you already tested it

Section 2, DFS methods, actually L2X is more of an explainability method than FS. There should be a clear distinction between the two; explainability/interpretability methods are post-training and do not prevent model overfitting, while FS does.

The DDS architecture is basically a hypernetwork. This is how this type of architecture, which predicts the weights of another network, is termed in the literature. Citations to hypernetworks are also missing.


How is the top-M procedure (eq. 4) differentiable?
Also, if you keep the top-M, why do you need the concrete layer and the regularization? This seems like two mechanisms pushing towards the same thing; why isn’t there an ablation demonstrating the contribution of each component?

The concrete gate is not properly described in Eqs. 3+5..
In the Hard Concrete distribution, there should be injected noise, without injected noise it’s not a concrete gate.
See Eq. 10 and 11 in Louizos et al. ICLR 2018.

In eq. 5 there is a sum with no index, what are you summing over?
How are all these hyperparameters set? And how is it possible that the regularization parameter is the same for all datasets?

Suddenly in Eq. 6 there is injected noise in the concrete gate, which is more correct but in contradiction to previous equations.

Commas missing after many equations

The decaying factor is not clearly motivated, and its effect is not demonstrated.

P5. near the end, “more complex networks that”-- should be “ than”?

Section 4.1.1 “than” should be “as”

It is unclear how the U net is actually reducing the dimension of the data, and how exactly the authors compare the number of selected features by DDS to the UNET?
Also, for the memory plot, do you count the weights of both networks in the DDS and compare them to the weights of the UNET?

P8. “thos” should be “those”

“It is like those pixels were not selected” not properly phrased.

All baselines used for clustering are actually self-supervised schemes that require image augmentations. So, this evaluation is not really unsupervised in the sense that it is very specific to image domain. Which again is less relevant for feature selection.

The main benefit of unsupervised feature selection is that it can actually improve clustering capabilities, but this is the case for high dimensional low sample data, and typically not images.
See examples:

Solorio-Fernández, Saúl, J. Ariel Carrasco-Ochoa, and José Fco Martínez-Trinidad. "A review of unsupervised feature selection methods." Artificial Intelligence Review 53.2 (2020): 907-948.

Mitra, Pabitra, C. A. Murthy, and Sankar K. Pal. "Unsupervised feature selection using feature similarity." IEEE transactions on pattern analysis and machine intelligence 24.3 (2002): 301-312.

He, Xiaofei, Deng Cai, and Partha Niyogi. "Laplacian score for feature selection." Advances in neural information processing systems 18 (2005).

Lindenbaum, Ofir, et al. "Differentiable unsupervised feature selection based on a gated laplacian." Advances in Neural Information Processing Systems 34 (2021): 1530-1542.



All focus on tabular data and improve clustering performance.

---

### Meta-Review · Area_Chair_LvJP · 2023-12-06

**Metareview:**

The paper advocates feature selection as opposed to feature extraction/engineering to achieve data compression and predictive learning, particularly in the context of dynamic environment.

The reviewers  pointed at quite a few references, asking the authors to position their contributions w.r.t. these related work.
The empirical validation should  preferably consider tabular data (as opposed to MNIST) to evidence the merits of the approach.
The reproducibility of the approach (adjustment of hyper-parameters) is insufficient.
The writing of the paper is not up to the ICLR standards.

The authors did not reply: it seems that there is a general consensus about this paper being premature for publication.

**Justification For Why Not Higher Score:**

N/A

**Justification For Why Not Lower Score:**

N/A

---

### Decision · Program_Chairs · 2024-01-16

Reject